# Identifying Root Causes of Important Service Failures across Medical Examination Processes with Integration of 4M1E, ITLV, GRA, DEMATEL and FMEA

**DOI:** 10.3390/healthcare10112283

**Published:** 2022-11-14

**Authors:** Hongying Fei, Yanmei Zhu, Yiming Kang, Suxia Shi, Xueguo Xu

**Affiliations:** 1School of Management, Shanghai University, 99 Shangda Road, Shanghai 200444, China; 2Shanghai Pulmonary Hospital, 507 Zhengmin Road, Shanghai 200433, China

**Keywords:** service quality, medical examination process, causes of failure, FMEA, DEMATEL

## Abstract

Medical examination plays an essential role in most medical treatment processes, and thus, the quality of service relevant to medical examination has great impact on patient satisfaction. The targeted hospital has long been faced with the problem that patient satisfaction of its medical examination department is below average. An assessment model, integrating 4M1E, ITLV, GRA, DEMATEL and FMEA, was developed in this study to identify the root causes of important service failures across medical examination processes, where (1) a cause-and-effect diagram was enhanced with 4M1E, identifying the list of failure modes relevant to service quality over the medical examination process with the 4M1E analysis framework, (2) FMEA experts were enabled to report their assessment results in their preferred ways by using the ITLV scheme, (3) causes of failure to failure modes with was figured out with DEMATEL, and (4) the evaluation results were improved by integrating GRA. Experimental results obtained by the proposed approach are compared with several benchmarks, and it was observed that (1) the results obtained by the proposed model are more suitable when FMEA experts prefer using different assessment languages versus other approaches; (2) the proposed model can figure out the key root causes according to their impact on overall failure modes.

## 1. Introduction

Medical examination plays an essential part in the majority of medical treatment processes, and thus, the quality of service relevant to medical examination processes has great impact on patient satisfaction. Shanghai Pulmonary Hospital, a medium-size specialized hospital located in Shanghai, China, has long been faced with the problem that patient satisfaction of its medical examination department is below average, according to patient satisfaction reports of this hospital. Considering that more and more hospitals have regarded patient satisfaction as the most important indicator measuring service quality in recent years [1], it is essential to reveal the key factors resulting in nonsatisfaction of patients and to give valuable suggestions for improving service quality for the targeted medical examination department. 

According to the literature, it was observed that there are few research studies on improving the service quality of medical examination, and most studies focus on either specific problems or certain medical examination activities such as the performance of C-arm X-ray machines, service assessments of radiographers’ experiences relevant to patient safety incidents, etc. [1,2,3,4,5]. However, few studies assess the service quality relevant to medical examination from the perspective of the entire process, i.e., from the time the patient makes an appointment for examination to receiving the final examination report. 

When the literature review was extended to include studies concerning service quality improvement for organizations other than the medical examination department, it was observed that many hybrid assessment models have been developed to draw on each other’s strengths. For example, Raziei et al. (2018) combined SERVQUAL with group decision making and QFD [6], Serkan and Semih (2019) integrated SERVQUAL and QFD with FMEA [7], and Tuzkaya et al. (2019) combined IVIF and PROMETHEE to overcome service quality problems from the perspectives of patients [8]. The ITLV scheme (2-tuple linguistic variables) has been successfully combined by some researchers, such as Li and He (2021), with assessment models to reduce both the uncertainty of evaluation information and the diversity of evaluation results [9]. 

Furthermore, it was observed that fuzzy techniques are widely applied in service quality assessment to deal with uncertainties. For instance, Liu et al. (2019) enhanced the performance of a fuzzy multi-attribute decision-making model by using grey relational analysis (GRA) when evaluating hospital service performance [10]. Yucesan and Gul (2020) combined Pythagorean FAHP and Pythagorean fuzzy technique to provide an accurate decision-making process for evaluating hospital service quality [11]. Alkafaji and Al-Shamery (2020) improved a service quality evaluation model by using fuzzy inference that can assess the quality of service in a similar way to the human experts of healthcare [12]. Akram et al. (2022) proposed a decision-making approach retaining the fascinating traits of the conventional VIKOR method in the context of the enrich multidimensional complex Fermatean fuzzy N-soft set [13]. Akram et al. (2022) developed a structure by the fusion of Fermatean fuzzy sets and linguistic term sets to deal with decision-making problems involving qualitative information [14]. Yüksel and Dinçer (2022) utilized hesitant 2-tuple interval-valued Pythagorean fuzzy DEMATEL to enhance the performance of the evaluation model [15].

Among the existing approaches, observed in the literature, developed for service quality improvement, it can be observed that FMEA (failure mode and effect analysis) and its variants have been widely used. A further literature review showed that FMEA and its variants can help prevent possible failure during service delivery in various fields from the perspectives of the entire process. For instance, Najafpour et al. (2017) applied FMEA to improve the efficiency of the blood transfusion process in a teaching general hospital [16]. Sayyadi Tooranloo et al. (2018) utilized FMEA to evaluate knowledge management failure factors in an intuitionistic fuzzy environment [17]. Mete (2019) applied a variant of FMEA to assess occupational risks in a natural gas pipeline construction project [18]. Lo et al. (2020) applied FMEA to identify critical failure modes of individual equipment components or processes to improve the development of plans [19]. Zhou et al. (2021) developed a FMEA-based approach to facilitate risk analysis of product design under uncertainty [20]. Although the traditional FMEA model suffers from several drawbacks when assessing the risk prioritization of potential failure modes, many efforts have been made to enhance the performance of the FMEA model in service quality evaluation so that many enhanced FMEA models can be successfully applied in various field for service quality improvement [21,22,23,24,25,26].

Encouraged by the success of FMEA models, this study aims at developing an advanced FMEA model for improving the service quality of medical examination in the targeted hospital concerning patient satisfaction, from the perspective of the entire medical examination process, where the following approaches have been integrated with FMEA to improve the FMEA assessment results: (1) The cause-and-effect diagram enhanced by 4M1E (man, machine, material, method and environment) structure was applied to systematically identify both the list of potential failure modes (FMs) and that of failure causes (CFs). (2) The ITLV (2-tuple linguistic variables), a multi-grained language scheme, was utilized to improve the collection of assessment opinions from FMEA experts. (3) DEMATEL (decision-making trial and evaluation laboratory) was applied to reveal the relationships between CFs and FMs. (4) The GRA (grey relation analysis) method was adopted to address the loss and uncertainty of decision makers’ judgements. 

The rest of the paper is organized as follows: In Section 2, the theorical framework of our FMEA model is introduced. In Section 3, the case study of the targeted hospital is detailed. Discussions about this study are given in Section 4, and this paper concludes in Section 5. 

## 2. Methodology

### 2.1. General Procedure of the Proposed Assessment Model

The general procedure of the FMEA model, an advanced service quality assessment model developed in this study, is as follows:

**Step 1:** Confirming central theme

In this study, the FMEA model is developed to identify potential risks involving the quality of services relevant to the medical examination process in the targeted hospital.

**Step 2:** Establishing a team with authority and representativeness

Five professionals, including two technicians, two nurses and one report physician, who have been responsible for various job positions across the medical examination process in the targeted hospital over one year, are invited as FMEA experts.

**Step 3:** Drawing the flowchart

The list of activities involved in the examination process are identified based on full discussions with the invited FMEA experts, and the flowchart of process is confirmed by those FMEA experts as well.

**Step 4:** Identifying the list of failure modes, failure causes and effects of failure

After full discussions with the expert group, we identify not only the failure modes and their possible effect, but also failure causes with the cause-and-effect diagram enhanced by the 4M1E structure. 

**Step 5:** Scoring and computing the RPN

First, the ITLV (interval 2-tuple linguistic variables) is applied to integrate the evaluation data collected from FMEA experts with multi-grained language. 

Then, GRA is used to ranking FMs (failure modes) according to their RPNs.

Afterward, DEMATEL is applied to identify the root causes for each failure mode.

**Step 6:** Devising and carrying out solutions

Based on the results obtained in Step 5, suggestions are given to improve the service quality across the medical examination process in the targeted hospital.

For better understanding, the contribution of the hybridization mechanism to improve the performance of traditional FMEA is depicted in Figure 1, and details about how each component works are given in the rest of this section.

### 2.2. Linguistic Variables

#### 2.2.1. 2-Tuple Linguistic Variables

2-tuple linguistic variables were firstly identified by Herrera and Martinez based on the concept of symbol translation [1]. In this study, applied is the generalized 2-tuple linguistic variables model raised by Chen and Tai in 2005 based on the concept of 2-tuple linguistic variables to deal with and compare linguistic variables from linguistic evaluation sets with different granularity [27]. Here below are some definitions relevant to 2-tuple linguistic variables.

**Definition** **1.**
*Let $S=\left\{S_{0},S_{1},\ldots ,S_{g}\right\}$ be a linguistic term set and $\beta \in \left[0,1\right]$ be a value representing the result of standardization. Then, the generalized translation function Δ used to obtain the 2-tuple linguistic variable equivalent to β can be defined as follows [1]:*


(1)$$\Delta :\left[0,1\right]\to S\times \left[-\frac{1}{2g},\left.\frac{1}{2g}\right)\right.\;$$(2)$$\Delta \left(\beta \right)=\left(S_{i},\alpha \right),with\;\left\{\begin{array}{c} S_{i},i=round\left(\beta \cdot g\right)\\ \alpha =\beta -\frac{i}{g},\alpha \in \left[-\frac{1}{2g},\left.\frac{1}{2g}\right)\right. \end{array}\right.$$where:(a)round means rounding off operation;(b)· means normal multiplication;(c)$S_{i}$ is the closet linguistic evaluation to β;(d)α is the value of the symbolic translation depending on *g*;(e)g is linked with the number of linguistic terms in *S*. For example, there are 5 linguistic terms in *S*; then, g=4 and $\alpha \in \left[-0.125,\left.0.125\right)\right.$

**Definition** **2.**
*Let $S=\left\{S_{0},S_{1},\ldots ,S_{g}\right\}$ be a linguistic term set and $\left(S_{i},\alpha \right)$ be a 2-tuple. The function $\Delta^{-1}$ that can translate a 2-tuple linguistic variable to its equivalent numerical value $\beta \in \left[0,1\right]$, which can be used to compare different multi-granularity linguistic terms, can be defined as follows:*



(3)
$$\Delta^{-1}:\;S\times \left[-\frac{1}{2g},\left.\frac{1}{2g}\right)\to \left[0,1\right]\right.\;$$



(4)
$$\Delta^{-1}\left(S_{i},\alpha \right)=\frac{i}{g}+\alpha =\beta$$


**Definition** **3.**
*Let $\left(S_{k},\alpha_{1}\right)$ and $\left(S_{l},\alpha_{2}\right)$ be two 2-tuples, and define the following rules:*
If k<l, then $\left(S_{k},\alpha_{1}\right)<\left(S_{l},\alpha_{2}\right)$;If k=l, then (1)If $\alpha_{1}=\alpha_{2}$, then $\left(S_{k},\alpha_{1}\right)=\left(S_{l},\alpha_{2}\right)$;(2)If $\alpha_{1}<\alpha_{2}$, then $\left(S_{k},\alpha_{1}\right)<\left(S_{l},\alpha_{2}\right)$;(3)If $\alpha_{1}>\alpha_{2}$, then $\left(S_{k},\alpha_{1}\right)>\left(S_{l},\alpha_{2}\right)$.


In the operation of 2-tuple linguistic variables, function Δ and function $\Delta^{-1}$ ensure no information loss during translation.

**Definition** **4.**
*Let $X=\left\{\left(r_{1},\alpha_{1}\right),\left(r_{2},\alpha_{2}\right),\ldots ,\left(r_{n},\alpha_{n}\right)\right\}$ be a 2-tuple set and $W={\left(W_{1},W_{2},\ldots ,W_{n}\right)}^{T}$ be their associated weights, with $W_{i}\in \left[0,1\right],i=1,2,\ldots ,n$, $\overset{n}{\underset{i=1}{\sum }}W_{i}=1$. The 2-tuple weighted average (TWA) is defined as*



(5)
$$TWA\left(X\right)=\Delta \left(\frac{1}{n}\sum_{i=1}^{n}W_{i}\Delta^{-1}\left(r_{i},\alpha_{i}\right)\right)=\Delta \left(\frac{1}{n}\sum_{i=1}^{n}W_{i}\beta_{i}\right)\;$$


#### 2.2.2. Interval 2-Tuple Linguistic Variables (ITLV)

**Definition** **5.**
*Let $S=\left\{S_{0},S_{1},\ldots ,S_{g}\right\}$ be a linguistic term set. An interval 2-tuple linguistic variable is composed of two 2-tuples, denoted by $\left[\left(S_{i},\alpha_{1}\right),\left(S_{j},\alpha_{2}\right)\right]$, with i<j and $\alpha_{1}<\alpha_{2}$. $S_{i}\left(S_{j}\right)$ and $\alpha_{1}\left(\alpha_{2}\right)$ represent the linguistic label of the predefined linguistic term set S and symbolic translation.*


Interval 2-tuple linguistic variable has the same meaning with $\left[\beta_{1},\beta_{2}\right]$ ($\beta_{1}$,$\;\beta_{2}\in \left[0,1\right],$ $\beta_{1}\leq \;\beta_{2}$) and is derived by the following function:(6)$$\Delta \left[\beta_{1},\beta_{2}\right]=\left[\left(S_{i},\alpha_{1}\right),\left(S_{j},\alpha_{2}\right)\right]\;with\;\left\{\begin{array}{c} S_{i},\;i=round\left(\beta_{1}\cdot g\right)\\ S_{j},\;j=round\left(\beta_{2}\cdot g\right)\\ \alpha_{1}=\beta_{1}-\frac{i}{g},\alpha_{1}\in \left[-\frac{1}{2g},\left.\frac{1}{2g}\right)\right.\\ \alpha_{2}=\beta_{2}-\frac{i}{g},\alpha_{2}\in \left[-\frac{1}{2g},\left.\frac{1}{2g}\right)\right. \end{array}\right.$$

On the contrary, there always exists function $\Delta^{-1}$, which can convert $\left[\left(S_{i},\alpha_{1}\right),\left(S_{j},\alpha_{2}\right)\right]$ into $\left[\beta_{1},\beta_{2}\right]$($\beta_{1}$,$\;\beta_{2}\in \left[0,1\right],$ $\beta_{1}\leq \;\beta_{2}$). Function $\Delta^{-1}$ is defined as follows:(7)$$\Delta^{-1}\left[\left(S_{i},\alpha_{1}\right),\left(S_{j},\alpha_{2}\right)\right]=\left[\frac{i}{g}+\alpha_{1},\frac{i}{g}+\alpha_{2}\right]=\left[\beta_{1},\beta_{2}\right]\;$$
when $S_{i}=S_{j}$ and $\alpha_{1}=\alpha_{2}$, the interval 2-tuple linguistic variable can be simplified as a 2-tuple one.

**Definition** **6.**
*Let $\tilde{X}=\left\{\left[\left(r_{1},\alpha_{1}\right),\left(t_{1},\varepsilon_{1}\right)\right],\left[\left(r_{2},\alpha_{2}\right),\left(t_{2},\varepsilon_{2}\right)\right],\ldots ,\left[\left(r_{n},\alpha_{n}\right),\left(t_{n},\varepsilon_{n}\right)\right]\right\}$ be an interval 2-tuple set, and W = ${\left(W_{1},W_{2},\ldots ,W_{n}\right)}^{T}$ be their associated weights, with $W_{i}\in \left[0,1\right],i=1,2,\ldots ,n$, $\overset{n}{\underset{i=1}{\sum }}W_{i}=1$. Then, the interval 2-tuple weighted average (ITWA) is defined as*



(8)
$$ITWA\left(\tilde{X}\right)=\Delta \left[\sum_{i=1}^{n}W_{i}\Delta^{-1}\left(r_{i},\alpha_{i}\right),\sum_{i=1}^{n}W_{i}\Delta^{-1}\left(t_{i},\varepsilon_{i}\right)\right]$$


**Definition** **7.**
*Let $\tilde{a}=\left[\left(r_{1},\alpha_{1}\right),\left(t_{1},\varepsilon_{1}\right)\right]$ and $\tilde{b}=\left[\left(r_{2},\alpha_{2}\right),\left(t_{2},\varepsilon_{2}\right)\right]$ be two interval 2-tuples, then the normalized Euclidean distance between $\tilde{a}$ and $\tilde{b}$ is defined as following:*



(9)
$$d\left(\tilde{a},\tilde{b}\right)=\Delta \sqrt{\left[{\left(\Delta^{-1}\left(r_{1},\alpha_{1}\right)-\Delta^{-1}\left(r_{2},\alpha_{2}\right)\right)}^{2}+{\left(\Delta^{-1}\left(t_{1},\varepsilon_{1}\right)-\Delta^{-1}\left(t_{2},\varepsilon_{2}\right)\right)}^{2}\right]}$$


### 2.3. Grey Relation Analysis (GRA)

Grey theory was proposed in 1982 and is widely used in situations with multiple input, imperfection and uncertain information, especially in [28]. Grey theory is composed of 6 main research methods, and GRA is one of the most important of them.

Supposing that $X=\left\{X_{0},X_{1},\ldots ,X_{m}\right\}$ is a grey relational factor set, $X_{0}\in X$ represents the reference sequence and $X_{i}$, i=1,2,…,m, represents the comparative sequence. $X_{0}$ and $X_{i}$ contain *n* elements, indicated as $X_{0}=\left(x_{0}\left(1\right),x_{0}\left(2\right),\ldots ,x_{0}\left(k\right),\ldots ,x_{0}\left(n\right)\right)$, $X_{i}=\left(x_{i}\left(1\right),x_{i}\left(2\right),\ldots ,x_{i}\left(k\right),\ldots ,x_{i}\left(n\right)\right)$, i=1,2,…,n, k=1,2,…,n.

$x_{0}\left(k\right)$ and $x_{i}\left(k\right)$ are the *k*th elements of reference and comparative sequence, and their grey relation coefficient can be calculated through the following equation:(10)$$\gamma \left(x_{0}\left(k\right),x_{i}\left(k\right)\right)=\frac{\underset{i}{\min}\;\underset{k}{\min}\left|x_{0}\left(k\right)-x_{i}\left(k\right)\right|+\xi \;\underset{i}{\max}\;\underset{k}{\max}\left|x_{0}\left(k\right)-x_{i}\left(k\right)\right|}{\left|x_{0}\left(k\right)-x_{i}\left(k\right)\right|+\xi \;\underset{i}{\max}\;\underset{k}{\max}\left|x_{0}\left(k\right)-x_{i}\left(k\right)\right|}$$

*ξ* is the distinguishing coefficient, $\xi \in \left[0,1\right]$. ξ=0.5 is applied.

The grey relational degree between $x_{0}$ and $x_{i}$ is indicated by the following equation:(11)$$\varphi \left(x_{0},x_{i}\right)=\frac{1}{n}\sum_{k=1}^{n}\gamma \left(x_{0}\left(k\right),x_{i}\left(k\right)\right)$$

### 2.4. Decision-Making Trial and Evaluation Laboratory (DEMATEL)

DEMATEL was proposed by the Battelle Memorial Institute of the Geneva Research Center to resolve complex social issues through setting up a matrix to compute the direct and indirect relationships between elements [28].

DEMATEL can be divided into 5 steps as follows.

**Step 1:** List and define the factors in a complex system, and then, design a form to demonstrate their causal relationship.

**Step 2:** Establish an initial direct-relation matrix *X*. Value in the matrix represents the incidence between factors, obtained by pairwise comparisons in terms of influences and directions. $x_{ij}$ is denoted as the degree to which factor *i* affects factor *j*.
$$X=\left[\begin{array}{cccc} 0 & x_{12} & \ldots & x_{1n}\\ x_{21} & 0 & \ldots & x_{2n}\\ \vdots & \vdots & \ddots & \vdots \\ x_{n1} & x_{n2} & \cdots & 0 \end{array}\right]$$

**Step 3:** Normalize the initial direct-relation matrix *X*. The normalized direct-relation matrix (*N*) can be obtained by the following equation:(12)$$\lambda =\frac{1}{max_{1\leq i\leq n}\sum_{j=1}^{n}x_{ij}}$$
(13)N=λX

**Step 4:** Calculate the total relation matrix (*T*).
(14)$$T=\underset{k\to \infty }{\lim}\left(N+N^{2}+\cdots +N^{k}\right)=N{\left(I-N\right)}^{-1}$$

**Step 5:** Calculate the sum of every row and column in matrix *T*. Let $R_{i}$ be sum of the $i^{th}$ row and $C_{j}$ be sum of the $j^{th}\;$ column. $R_{j}$ and $C_{j}$ contain the direct and indirect relationships among factors.
(15)$$R_{j}=\sum_{j=1}^{n}t_{ij}\;\left(i=1,2,\ldots ,n\right)$$
(16)$$C_{j}=\sum_{i=1}^{n}t_{ij}\;\left(j=1,2,\ldots ,n\right)$$

### 2.5. GRA-DEMATEL-Based FMEA (GD-FMEA Method)

In this subsection, the GD-FMEA method is combined with the interval 2-tuple linguistic variables, where GRA and DEMATEL will be demonstrated.

Supposing that (a) an expert team is composed of l expert $\left(DM_{k},\;k=1,2,\ldots ,l\right)$, and $\lambda_{k}$ is their associated weights, in which $k=1,2,\ldots l,\;\lambda_{k}>0$, and $\sum_{k=1}^{l}\lambda_{k}=1$; (b), there are *m* causes of failure $\left(CF_{i},\;i=1,2,\ldots ,m\right)$ with *n* risk factors $\left(RF_{j},\;j=1,2,\ldots ,n\right)$. Weights represent the importance of each expert when they evaluate the risk. Every expert can use a different linguistic term set S ($S=\left\{S_{0},S_{1},\ldots ,S_{g}\right\}$) to evaluate CFs. Let $D_{k}={\left(d_{ij}^{k}\right)}_{m\times n}$ be the linguistic evaluation matrix of the *k*th expert, where $d_{ij}^{k}$ is the linguistic assessment provided by $DM_{k}$ on the assessment of $CF_{i}$, with respect to $RF_{j}$. Let $w_{j}^{k}$ be the linguistic weight of risk factor $RF_{j}$ given by $DM_{k}$ to reflect its relative importance in the determination of risk priorities of causes of failure and failure modes. Based on the previously mentioned assumptions, the GD-FMEA method can be divided into the following steps:

**Step 1:** Establish a team with authority and representativeness, and then, list flowchart, failure modes, causes and effect of failure.

**Step 2:** Experts use different linguistic term sets to evaluate the risk factors and CFs; then, convert the linguistic decision matrix $D_{k}={\left(d_{ij}^{k}\right)}_{m\times n}$ into an interval 2-tuple linguistic decision matrix $\tilde{R}={\left({\tilde{r}}_{ij}^{k}\right)}_{m\times n}={\left(\left[\left(r_{ij}^{k},0\right),\;\left(t_{ij}^{k},0\right)\right]\right)}_{m\times n}$, where $r_{ij}^{k}$, $t_{ij}^{k}\in S,$ $S=\left\{S_{0},S_{1},\ldots ,S_{g}\right\}$ and $r_{ij}^{k}<t_{ij}^{k}$.

**Step 3:** Aggregate the experts’ opinions to construct a collective interval 2-tuple linguistic decision matrix $\tilde{R}={\left({\tilde{r}}_{ij}\right)}_{m\times n}$ and obtain the aggregated 2-tuple linguistic weight of each risk factor $\omega ={\left[\left(\omega_{j},\alpha_{\omega j}\right)\right]}_{1\times n}$, where
(17)$${\tilde{R}}_{ij}=\left[\left(r_{ij},\alpha_{ij}\right),\left(t_{ij},\varepsilon_{ij}\right)\right]=ITWA{\left({\tilde{r}}_{ij}^{k}\right)}_{m\times n}\;=(\left[(r_{ij}^{1},0),\;(t_{ij}^{1},0)],\;[(r_{ij}^{2},0),\;(t_{ij}^{2},0)],\ldots ,\;[(r_{ij}^{l},0),\;(t_{ij}^{l},0)\right])\;=\Delta \left[\sum_{k=1}^{l}\lambda_{k}\Delta^{-1}\left(r_{xij}^{k},0\right),\sum_{k=1}^{l}\lambda_{k}\Delta^{-1}\left(t_{ij}^{k},0\right)\right],\;i=1,2,\ldots m,j=1,2,\ldots ,n$$
(18)$$\left(\omega_{j},\alpha_{\omega j}\right)=\left[TWA\left(\omega_{j}^{1},0\right),\left(\omega_{j}^{2},0\right),\ldots ,\left(\omega_{j}^{l},0\right)\right]=\Delta \left[\sum_{k=1}^{l}\lambda_{k}\Delta^{-1}\left(\omega_{j}^{k},0\right)\right]$$

**Step 4:** Calculate the weight of risk factors (S, O, D)

Based on the aggregated weights of risk factors ${\left(\omega_{j},\alpha_{\omega j}\right)}_{1\times n}$, the normalized risk factor weights can be obtained as follows:(19)$$\bar{\omega_{j}}=\frac{\Delta^{-1}\left(\omega_{j},\alpha_{\omega j}\right)}{\sum_{j=1}^{n}\Delta^{-1}\left(\omega_{j},\alpha_{\omega j}\right)},j=1,2,\ldots ,n$$

**Step 5:** This step is related to GRA; thus, it can be subdivided into four steps:(1)Determine the reference sequence $r_{0}$
(20)$$r_{0}={\left(r_{ij}\right)}_{1\times n}=\left[\left(s_{0},0\right),\left(s_{0},0\right),\ldots ,\left(s_{0},0\right)\right]$$(2)Calculate the distances (differences) between the comparative sequences and the reference one, and establish the distance matrix $D_{0}={\left[d\left({\tilde{r}}_{ij},r_{0j}\right)\right]}_{m\times n}$
(21)$$D\left({\tilde{r}}_{ij},r_{0j}\right)=\Delta \sqrt{\frac{1}{2}\left[{\left(\Delta^{-1}\left(r_{ij},\alpha_{ij}\right)-\Delta^{-1}\left(s_{0},0\right)\right)}^{2}+{\left(\Delta^{-1}\left(t_{ij},\varepsilon_{ij}\right)-\Delta^{-1}\left(s_{0},0\right)\right)}^{2}\right]}$$(3)Calculate the grey relational coefficient $\gamma_{ij}$The grey relational coefficient represents how close ${\tilde{r}}_{ij}$ is to $r_{0j}$. The larger the grey relational coefficient is, the closer ${\tilde{r}}_{ij}$ and $r_{0j}$ are:(22)$$\gamma_{ij}=\Delta \left(\frac{\Delta^{-1}\delta_{min}+\zeta \Delta^{-1}\delta_{max}}{\Delta^{-1}\delta_{ij}+\zeta \Delta^{-1}\delta_{max}}\right),\;i=1,2,\ldots ,m,j=1,2,\ldots ,n$$
where $\delta_{ij}=d\left({\tilde{r}}_{xij},r_{xij}\right)$, $\delta_{min}=\underset{xi}{min}\underset{j}{min}\delta_{ij}$, $\delta_{max}=\underset{xi}{max}\underset{j}{max}\delta_{ij}$, the distinguishing coefficient $\xi \in \left[0,1\right]$. ξ=0.5 is applied.(4)Estimate the grey relational degree $\varphi_{i}$
(23)$$\varphi_{i}=\Delta \left(\sum_{j=1}^{n}\bar{\omega_{j}}\Delta^{-1}r_{ij}\right)$$

**Step 6:** Construct the initial relation matrix *Y* based on the result of Step 5.

**Step 7:** Normalize matrix *Y* to obtain the normalized relation matrix *H*. This step can be subdivided into two steps.

(1)Calculate the sum of every row and column and use Equation (12) to obtain the reciprocal of the maximum of summed columns and rows.(2)Use Equation (13) to obtain the normalized relation matrix *H*.

**Step 8:** Calculate the direct and indirect relative severity matrix *T*.

Use Equation (14) to obtain the direct and indirect severity matrix *T*.

**Step 9:** Calculate the value of (R + C) and (R − C) based on Equation (15) and Equation (16).

(R + C) represents the total relationships between the cause and effect of specific criteria.

(R − C) represents influence, the differences between the cause and effect of specific criteria [29].

## 3. Case Study

In this section, it will be demonstrated how the GD-FMEA is applied to derive the root causes of failures resulting in poor service quality across the medical examination process in the targeted hospital located in Shanghai, China, and the steps are as follows. 

**Step 1:** Define the list of failure modes (FMs) and corresponding causes of failure (CFs). 

As shown in Figure 2, the process relevant to a medical examination in the targeted hospital consists of three stages, where the pre-inspection process starts from the registration of the patient until the arrival of the patient at the medical technology department at the appointment time; the per-inspection process consists of all the relevant activities carried out during the inspection; and the post-inspection process refers to obtaining the examination report after the inspection.

Based on literature review and after full discussion with the FMEA experts, the list of FMs defined in this study is constructed as shown in Table 1, where the corresponding CFs are identified with the cause-and-effect diagram enhanced by the 4M1E mechanism, as shown Figure 3, for FM_1_, as an example. 

**Step 2:** Collect individual assessment results from FMEA experts.

To ensure the quality of assessment data, the FMEA expert team is composed of five experts, called decision makers (DMs) hereafter, who are senior professionals invited from the medical technology department in the targeted hospital, including two technicians responsible for medical examination operations, two nurses for on-site services at the targeted medical examination department, and one report physician. Considering the job responsibilities of these experts can be regarded as having equally important impact on the satisfaction of the patient partaking in the medical examination; their respective weights are set the same, i.e., 0.2 for each. 

Furthermore, when collecting individual assessment data from the FMEA team, three linguistic term sets, whose structures are shown in Table 2, are provided in the questionnaires to enable the experts to fill in the questionnaires using their favorite rating system. 

According to the questionnaires collected from the FMEA experts, noted as DM_1_ to DM_5_, it was observed that DM_1_ preferred set C, DM_4_ selected set B, and the other three experts, i.e., DM_2_, DM_3_ and DM_5_, used set A.

As for the collection of experts’ opinions on the importance of risk factors, we applied a five-granularity linguistic term set D = {d_0_ = very unimportant (VU), d_1_ = unimportant (U), d_2_ = medium(M), d_3_ = important (I), d_4_ = very important (VI)}.

Since data collected from FMEA experts are multi-linguistic, they are firstly converted uniformly into interval 2-tuple linguistic variables. Table 3, Table 4 and Table 5 show the 2-tuple linguistic variables corresponding to Severity (S), Occurrence (O), Detection (D).

Interval 2-tuple linguistic variables for the importance of risk factors are shown in Table 6. 

**Step 3:** Construct an aggregated interval 2-tuple linguistic decision matrix.

Use Equation (17) to construct the aggregated interval 2-tuple linguistic decision matrix $\tilde{R}={\left({\tilde{r}}_{ij}\right)}_{m\times n}$, and use Equation (18) to obtain the aggregated 2-tuple linguistic weight of each risk factor $\omega ={\left[\left(\omega_{j},\alpha_{\omega j}\right)\right]}_{1\times n}$, as shown in Table 7.

**Step 4:** Normalize the weights of risk factors.

Use Equation (19) to normalize the weights of risk factors, as shown in the last line of Table 7.

**Step 5:** Set reference sequence. 

Use Equation (20) to determine the following reference sequence.
$$r_{0}=\left[\left(a_{4},0\right),\left(a_{4},0\right),\left(a_{4},0\right)\right]$$

**Step 6:** Establish distance matrix. 

Use Equation (21) to calculate the distances (differences) between the comparative sequences and the reference one, and then, establish the distance matrix $D_{0}={\left[d\left({\tilde{r}}_{ij},r_{0j}\right)\right]}_{m\times n}$, as shown in Table 8.

**Step 7:** Calculate grey relational coefficients. 

Determine the grey relational coefficient $\gamma_{ij}$ by using Equation (22), of which the results are shown in Table 9.

**Step 8:** Estimate grey relational degrees. 

Determine the grey relational degree $\varphi_{i}$ by using Equation (23), of which the results are shown in Table 10.

**Step 9:** Construct initial relation matrix among FMs and CFs. 

Use the grey relational degrees obtained in Step 8 to construct the initial relation matrix *Y* among FMs and CFs, as shown in Table 11.

**Step 10:** Construct direct and indirect relation matrix among FMs and CFs. 

Use Equation (13) and Equation (14) to obtain the direct and indirect relation matrix *T* among FMs and CFs, as shown in Table 12.

**Step 11:** Calculate the direct and indirect relative relationships.

Use Equation (15) and Equation (16) to calculate R for CFs and C for FMs, as shown in Table 13 and Table 14, respectively. 

**Step 12:** Rank the priority of FMs and CFs. 

Since the worst case is set as the reference sequence, it is reasonable to conclude that the bigger the value of R, the more significant a CF is, and similarly, the bigger C is, the higher priority FM must have. According to the results shown in Table 13, it can be observed that CF_22_, CF_16_, CF_4_, CF_3_, and CF_5_, whose values are close to 1, are considered as significant. With the results shown in Table 14, FM_5_, FM_1_ and FM_7_ are ranked as the top three failures modes resulting in poor service quality at the targeted medical examination department.

Considering that some failure causes may have impact on more than one failure mode, it is also interesting to further analyze the overall impact of causes to those three critical failure modes: FM_5_, FM_1_ and FM_7_. Considering the meanings of R and C, the priority of CFs relevant to these three critical FMs is ranked according to the descending order of (R + C), such that the bigger the value of (R + C) obtained, the higher priority a CF has, and more attention must be paid to improve the activities relevant to this CF. As shown in Table 15, it can be observed that the top five CFs have a firm relationship with the top three FMs and the top two CFs have great impact on FM_5_ and FM_7_. 

Furthermore, a comparison among the proposed GD-FMEA, an enhanced VIKOR (CFFS-VIKOR) [13], and a GRA-FMEA (G-FMEA) [21] is conducted to evaluate the performance of GD-FMEA. The results obtained by CFFS-VIKOR contain two parts, where CFFS-VIKOR-A corresponds to the assessment data collected with the five-granularity assessment language (Set A) and CFFS-VIKOR-C corresponds to the nine-granularity assessment language (set C). The results regarding the priorities of FMs and critical FM-related CFs are shown in Table 16 and Table 17, respectively.

## 4. Discussions

According to the results shown in Table 13, the FMs whose evaluation results (C) are significantly higher than the others are FM_5_ (C = 5.409, machines go wrong) and FM_1_ (C = 3.423, waiting too long in the queue for examination), where the former is related firmly with machine breakdown and the latter reveals the patients’ complaints about long waiting in the queue for examination. As for CFs, it can be observed that CF_22_ (software system of self-service reporting machine does not work), CF_16_ (patients do not use the self-service reporting machines according to the regulations), CF_4_ (report physician writes wrong patient’s name on report), CF_3_ (technicians missed certain inspection items or performed the wrong inspection) and CF_5_ (staff acts wrongly out of personal considerations) are ranked as the top five. Furthermore, their evaluation results (R) are quite close to 1, indicating that those causes have great impact on poor quality of relevant services in the targeted services. Further analysis of the relationship between these top FMs and CFs shows that the top two causes of failures (CF_22_ and CF_16_) firmly relate with FM_5_, which indicates that more and more patients derive the inspection report from the self-service reporting machine. Thus, not only the failures of the machine’s operating system but also improper operations of the machine may prevent the patients from successfully obtaining their inspection report, resulting in complaints from patients, and the hospital should not only ensure the normal status of the reporting machines but should also improve on-site assistance service for patients to derive inspection reports from the reporting machines. On the other hand, regarding the analysis of the significant causes related with FM_1_, i.e., CF_3_, CF_4_ and CF_5_, it can be concluded that all of them are human-related failures, although none of them relate to professionals, such as technicians and report physicians, but rather to nurses responsible for on-site reception and assistance services. Furthermore, it can be concluded that these three CFs are related to a lack of responsibility. Therefore, the suggestion for the targeted hospital is to organize a training program to improve the professionalism and responsibility of the staff, especially the examination technicians and the reporting physicians. 

According to the comparison regarding the priority of FMs as shown in Table 16, it can be observed that the results of GD-FMEA and G-FMEA are the same, and the top two FMs identified by these two approaches are the same as the CFFS-VIKOR model. As for the CFFS-VIKOR, we observed that the results are not consistent when different assessment language sets are applied. Furthermore, it can be observed that regarding the results shown in Table 17, the results obtained by GD-FMEA are consistent with those obtained by G-FMEA and CFFS-VIKOR-A for the top four CFs, and the same results can be obtained through G-FMEA for the top six CFs. It is reasonable to conclude that the proposed assessment model is much more stable than the approaches using mono-assessment language set and is consistent with the other assessment approaches. 

## 5. Conclusions

In this study, an assessment model, integrating 4M1E, ITLV, GRA, DEMATEL and FMEA, was developed to identify the root causes of important service failures across medical examination processes in a medium-sized specialty hospital, where (1) the cause-and-effect diagram was enhanced with 4M1E to identify the list of failure modes relevant to service quality over the medical examination process with 4M1E analysis framework, (2) FMEA experts were enabled to report their assessment results in their preferred ways by using ITLV scheme, (3) the causes of failure to failure modes were figured out with DEMATEL, and (4) the evaluation results were improved by integrating GRA. The case study was conducted in the targeted specialized hospital, which has long been faced with the problem that patient satisfaction of its medical examination department is below average. A comparison among the proposed GD-FMEA, CFFS-VIKOR and G-FMEA models was conducted to evaluate the performance of GD-FMEA. According to the comparison, the results obtained by GD-FMEA are consistent with those obtained by G-FMEA and CFFS-VIKOR-A for the top four CFs, and the same results were obtained as G-FMEA for the top six CFs. It is reasonable to conclude that the proposed assessment model was much more stable than approaches using mono-assessment language sets and was consistent with the other assessment approaches. CF_22_ (software system of self-service reporting machine does not work), CF_16_ (patients do not use the self-service reporting machines according to the regulations), CF_4_ (report physician writes wrong patient’s name on report), CF_3_ (technicians missed certain inspection items or performed the wrong inspection) and CF_5_ (staff acts wrongly out of personal considerations) were ranked as the top five. Therefore, several suggestions are presented as follows:Strengthen the daily maintenance of inspection machines and self-service machines, including software and hardware;Train volunteers to operate the self-service machines and assign them to instruct patients at the self-service machines;Establish a two-person inspection mechanism for reports to reduce the occurrence of errors.

## Figures and Tables

**Figure 1 healthcare-10-02283-f001:**
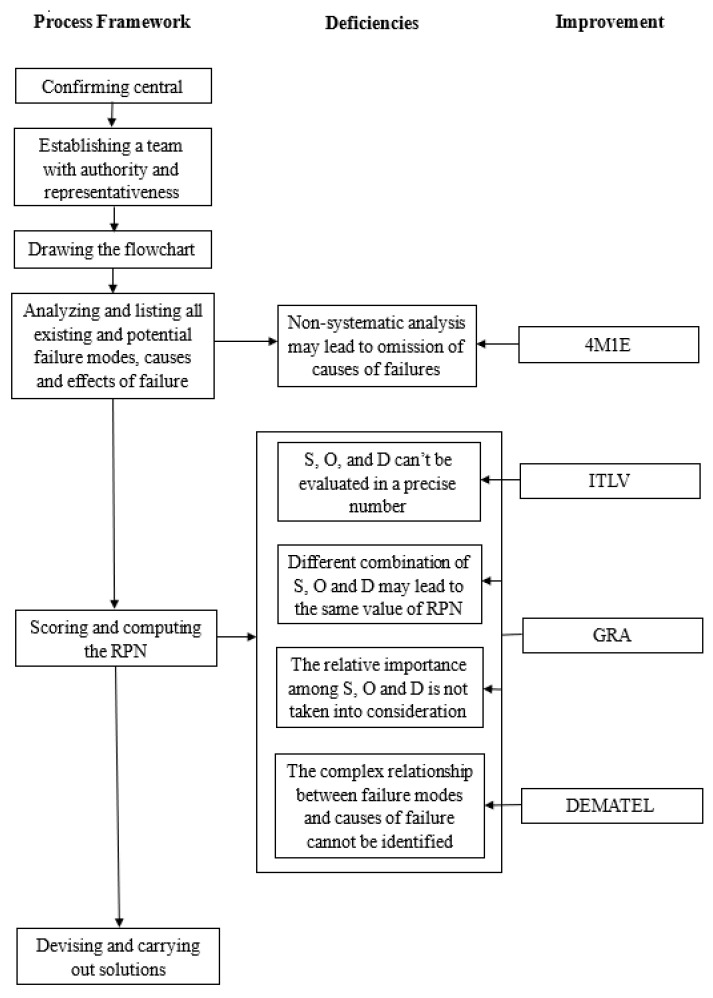
Comparison between the proposed FMEA and traditional one.

**Figure 2 healthcare-10-02283-f002:**
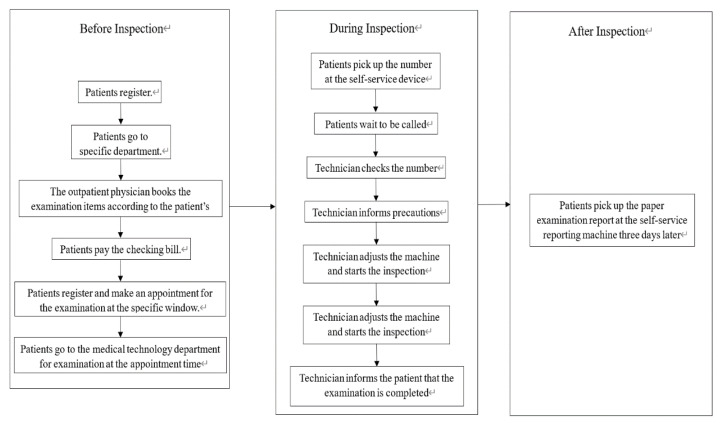
The flowchart of inspection.

**Figure 3 healthcare-10-02283-f003:**
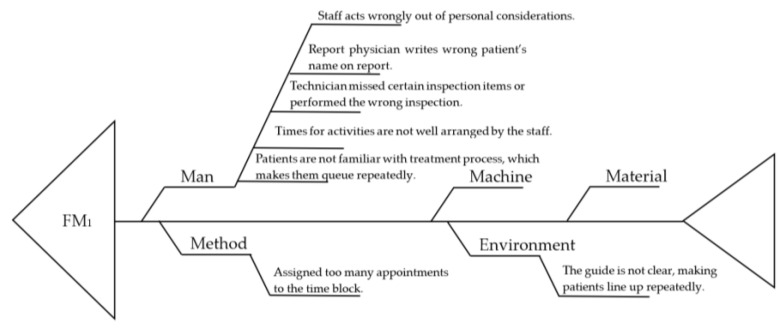
Cause-and-effect diagram enhanced by 4M1E scheme for identifying root causes of failure relevant to FM_1_.

**Table 1 healthcare-10-02283-t001:** List of FMs across the medical examination process in the targeted hospital.

FMs	Description of FM	Factors	CF	Description of CF
FM_1_	Waiting too long in the queue for an examination	Man	CF_1_	Patients are not familiar with treatment process, which makes them queue repeatedly
			CF_2_	Times for activities are not well arranged by the staff
			CF_3_	Technician missed certain inspection items or performed the wrong inspection
			CF_4_	Report physician writes wrong patient’s name on report
			CF_5_	Staff acts wrongly out of personal considerations
		Method	CF_6_	Assigned too many appointments to the time block
		Environment	CF_7_	The guide is not clear, making patients line up repeatedly
FM_2_	Poor management at reception	Man	CF_5_	Staff acts wrongly out of personal considerations
			CF_8_	Technicians do not verify the number, resulting in queue cutting
FM_3_	Patients do not appear on time	Environment	CF_9_	Room numbers are ambiguous
			CF_10_	Prompt tone sounds lightly
FM_4_	Checking part is not right	Man	CF_3_	Technician missed certain inspection items or performed the wrong inspection
			CF_11_	Patient and item names are not confirmed
			CF_12_	The outpatient physician does not book the examination items according to the patient’s needs
FM_5_	Machines go wrong	Man	CF_13_	Technical staff does not identify potential faults
			CF_14_	Technicians fail to operate the instrument as required
			CF_15_	Patients do not use the self-service number machine according to the regulations
			CF_16_	Patients do not use the self-service reporting machine according to the regulations
		Machine	CF_17_	Medical technical inspection equipment is not regularly maintained
			CF_18_	Self-service number machines are not regularly maintained
			CF_19_	Self-service reporting machines are not regularly maintained
			CF_20_	Software system of medical technical inspection equipment failures
			CF_21_	Software system of self-service number machine failures
			CF_22_	Software system of self-service reporting machine failures
		Environment	CF_23_	The instrument placement environment does not meet the requirements
FM_6_	Report does not tally with patient’s symptom	Man	CF_4_	Report physician writes wrong patient’s name on report
			CF_24_	Report physician inputs error of examination result
			CF_25_	Technician uninforms precautions
FM_7_	Patients do not fetch report on time	Man	CF_16_	Patients do not use the self-service reporting machine according to the regulations
		Machine	CF_22_	Software system of self-service reporting machine failures
			CF_26_	Self-service reporting machine goes wrong
		Material	CF_27_	Insufficient material such as paper and ink

**Table 2 healthcare-10-02283-t002:** Three linguistic term sets applied in this study (EL: Extremely Low; VL: Very Low; ML: Moderately Low; M: Moderate; MH: Moderately High; H: High; VH: Very High; EH: Extremely High).

Set	EL	VL	L	ML	M	MH	H	VH	EH
**A**		a_0_	a_1_		a_2_		a_3_	a_4_	
**B**		b_0_	b_1_	b_2_	b_3_	b_4_	b_5_	b_6_	
**C**	c_0_	c_1_	c_2_	c_3_	c_4_	c_5_	c_6_	c_7_	c_8_

**Table 3 healthcare-10-02283-t003:** Interval 2-tuple linguistic variables for Severity (S).

	DM_1_	DM_2_	DM_3_	DM_4_	DM_5_
FM_1_-CF_1_	[(c_0_,0),(c_0_,0)]	[(a_4_,0),(a_4_,0)]	[(a_3_,0),(a_3_,0)]	[(b_2_,0),(b_2_,0)]	[(a_1_,0),(a_1_,0)]
FM_1_-CF_2_	[(c_6_,0),(c_6_,0)]	[(a_1_,0),(a_1_,0)]	[(a_2_,0),(a_2_,0)]	[(b_0_,0),(b_0_,0)]	[(a_1_,0),(a_1_,0)]
FM_1_-CF_3_	[(c_6_,0),(c_6_,0)]	[(a_1_,0),(a_1_,0)]	[(a_0_,0),(a_0_,0)]	[(b_0_,0),(b_0_,0)]	[(a_0_,0),(a_0_,0)]
FM_1_-CF_4_	[(c_8_,0),(c_8_,0)]	[(a_1_,0),(a_1_,0)]	[(a_0_,0),(a_0_,0)]	[(b_0_,0),(b_0_,0)]	[(a_0_,0),(a_0_,0)]
FM_1_-CF_5_	[(c_4_,0),(c_4_,0)]	[(a_2_,0),(a_2_,0)]	[(a_0_,0),(a_0_,0)]	[(b_0_,0),(b_0_,0)]	[(a_1_,0),(a_1_,0)]
FM_1_-CF_6_	[(c_7_,0),(c_7_,0)]	[(a_0_,0),(a_0_,0)]	[(a_4_,0),(a_4_,0)]	[(b_1_,0),(b_1_,0)]	[(a_1_,0),(a_1_,0)]
FM_1_-CF_7_	[(c_5_,0),(c_5_,0)]	[(a_2_,0),(a_2_,0)]	[(a_3_,0),(a_3_,0)]	[(b_1_,0),(b_1_,0)]	[(a_1_,0),(a_1_,0)]
FM_2_-CF_5_	[(c_4_,0),(c_4_,0)]	[(a_1_,0),(a_1_,0)]	[(a_1_,0),(a_1_,0)]	[(b_0_,0),(b_0_,0)]	[(a_0_,0),(a_0_,0)]
FM_2_-CF_8_	[(c_6_,0),(c_6_,0)]	[(a_1_,0),(a_1_,0)]	[(a_1_,0),(a_1_,0)]	[(b_1_,0),(b_1_,0)]	[(a_0_,0),(a_0_,0)]
FM_3_-CF_9_	[(c_6_,0),(c_6_,0)]	[(a_2_,0),(a_2_,0)]	[(a_2_,0),(a_2_,0)]	[(b_2_,0),(b_2_,0)]	[(a_0_,0),(a_0_,0)]
FM_3_-CF_10_	[(c_6_,0),(c_6_,0)]	[(a_1_,0),(a_1_,0)]	[(a_2_,0),(a_2_,0)]	[(b_3_,0),(b_3_,0)]	[(a_0_,0),(a_0_,0)]
FM_4_-CF_3_	[(c_6_,0),(c_6_,0)]	[(a_1_,0),(a_1_,0)]	[(a_4_,0),(a_4_,0)]	[(b_2_,0),(b_2_,0)]	[(a_0_,0),(a_0_,0)]
FM_4_-CF_11_	[(c8,_0_),(c_8_,0)]	[(a_0_,0),(a_0_,0)]	[(a_4_,0),(a_4_,0)]	[(b_0_,0),(b_0_,0)]	[(a_0_,0),(a_0_,0)]
FM_4_-CF_12_	[(c_7_,0),(c_7_,0)]	[(a_0_,0),(a_0_,0)]	[(a_2_,0),(a_2_,0)]	[(b_0_,0),(b_0_,0)]	[(a_0_,0),(a_0_,0)]
FM_5_-CF_13_	[(c_6_,0),(c_6_,0)]	[(a_0_,0),(a_0_,0)]	[(a_4_,0),(a_4_,0)]	[(b_2_,0),(b_2_,0)]	[(a_1_,0),(a_1_,0)]
FM_5_-CF_14_	[(c_6_,0),(c_6_,0)]	[(a_3_,0),(a_3_,0)]	[(a_4_,0),(a_4_,0)]	[(b_3_,0),(b_3_,0)]	[(a_0_,0),(a_0_,0)]
FM_5_-CF_15_	[(c_4_,0),(c_4_,0)]	[(a_1_,0),(a_1_,0)]	[(a_2_,0),(a_2_,0)]	[(b_3_,0),(b_3_,0)]	[(a_1_,0),(a_1_,0)]
FM_5_-CF_16_	[(c_6_,0),(c_6_,0)]	[(a_1_,0),(a_1_,0)]	[(a_2_,0),(a_2_,0)]	[(b_3_,0),(b_3_,0)]	[(a_1_,0),(a_1_,0)]
FM_5_-CF_17_	[(c_4_,0),(c_4_,0)]	[(a_2_,0),(a_2_,0)]	[(a_4_,0),(a_4_,0)]	[(b_4_,0),(b_4_,0)]	[(a_1_,0),(a_1_,0)]
FM_5_-CF_18_	[(c_4_,0),(c_4_,0)]	[(a_2_,0),(a_2_,0)]	[(a_4_,0),(a_4_,0)]	[(b_4_,0),(b_4_,0)]	[(a_1_,0),(a_1_,0)]
FM_5_-CF_19_	[(c_4_,0),(c_4_,0)]	[(a_2_,0),(a_2_,0)]	[(a_2_,0),(a_2_,0)]	[(b_4_,0),(b_4_,0)]	[(a_1_,0),(a_1_,0)]
FM_5_-CF_20_		[(a_3_,0),(a_3_,0)]	[(a_3_,0),(a_3_,0)]	[(b_3_,0),(b_3_,0)]	[(a_0_,0),(a_0_,0)]
FM_5_-CF_21_	[(c_4_,0),(c_4_,0)]	[(a_3_,0),(a_3_,0)]	[(a_3_,0),(a_3_,0)]	[(b_3_,0),(b_3_,0)]	[(a_0_,0),(a_0_,0)]
FM_5_-CF_22_	[(c_6_,0),(c_6_,0)]	[(a_3_,0),(a_3_,0)]	[(a_3_,0),(a_3_,0)]	[(b_3_,0),(b_3_,0)]	[(a_0_,0),(a_0_,0)]
FM_5_-CF_23_	[(c_6_,0),(c_6_,0)]	[(a_1_,0),(a_1_,0)]	[(a_4_,0),(a_4_,0)]	[(b_1_,0),(b_1_,0)]	[(a_0_,0),(a_0_,0)]
FM_6_-CF_4_	[(c_7_,0),(c_7_,0)]	[(a_3_,0),(a_3_,0)]	[(a_4_,0),(a_4_,0)]	[(b_4_,0),(b_4_,0)]	[(a_0_,0),(a_0_,0)]
FM_6_-CF_24_		[(a_3_,0),(a_3_,0)]	[(a_4_,0),(a_4_,0)]	[(b_3_,0),(b_3_,0)]	[(a_0_,0),(a_0_,0)]
FM_6_-CF_25_	[(c_7_,0),(c_7_,0)]	[(a_1_,0),(a_1_,0)]	[(a_4_,0),(a_4_,0)]	[(b_1_,0),(b_1_,0)]	[(a_0_,0),(a_0_,0)]
FM_7_-CF_16_	[(c_6_,0),(c_6_,0)]	[(a_4_,0),(a_4_,0)]	[(a_2_,0),(a_2_,0)]	[(b_3_,0),(b_3_,0)]	[(a_0_,0),(a_0_,0)]
FM_7_-CF_22_	[(c_7_,0),(c_7_,0)]	[(a_4_,0),(a_4_,0)]	[(a_2_,0),(a_2_,0)]	[(b_2_,0),(b_2_,0)]	[(a_0_,0),(a_0_,0)]
FM_7_-CF_26_	[(c_6_,0),(c_6_,0)]	[(a_3_,0),(a_3_,0)]	[(a_2_,0),(a_2_,0)]	[(b_2_,0),(b_2_,0)]	[(a_0_,0),(a_0_,0)]
FM_7_-CF_27_	[(c_4_,0),(c_4_,0)]	[(a_1_,0),(a_1_,0)]	[(a_2_,0),(a_2_,0)]	[(b_2_,0),(b_2_,0)]	[(a_0_,0),(a_0_,0)]

**Table 4 healthcare-10-02283-t004:** Interval 2-tuple linguistic variables for Occurrence (O).

	DM_1_	DM_2_	DM_3_	DM_4_	DM_5_
FM_1_-CF_1_	[(c_3_,0),(c_3_,0)]	[(a_4_,0),(a_4_,0)]	[(a_3_,0),(a_3_,0)]	[(b_2_,0),(b_2_,0)]	[(a_1_,0),(a_1_,0)]
FM_1_-CF_2_	[(c_1_,0),(c_1_,0)]	[(a_2_,0),(a_2_,0)]	[(a_3_,0),(a_3_,0)]	[(b_2_,0),(b_2_,0)]	[(a_0_,0),(a_0_,0)]
FM_1_-CF_3_	[(c_6_,0),(c_6_,0)]	[(a_0_,0),(a_0_,0)]	[(a_4_,0),(a_4_,0)]	[(b_2_,0),(b_2_,0)]	[(a_0_,0),(a_0_,0)]
FM_1_-CF_4_	[(c_8_,0),(c_8_,0)]	[(a_2_,0),(a_2_,0)]	[(a_4_,0),(a_4_,0)]	[(b_1_,0),(b_1_,0)]	[(a_0_,0),(a_0_,0)]
FM_1_-CF_5_	[(c_4_,0),(c_4_,0)]	[(a_1_,0),(a_1_,0)]	[(a_2_,0),(a_2_,0)]	[(b_1_,0),(b_1_,0)]	[(a_0_,0),(a_0_,0)]
FM_1_-CF_6_	[(c_7_,0),(c_7_,0)]	[(a_0_,0),(a_0_,0)]	[(a_3_,0),(a_3_,0)]	[(b_1_,0),(b_1_,0)]	[(a_0_,0),(a_0_,0)]
FM_1_-CF_7_	[(c_5_,0),(c_5_,0)]	[(a_1_,0),(a_1_,0)]	[(a_3_,0),(a_3_,0)]	[(b_2_,0),(b_2_,0)]	[(a_0_,0),(a_0_,0)]
FM_2_-CF_5_	[(c_4_,0),(c_4_,0)]	[(a_1_,0),(a_1_,0)]	[(a_2_,0),(a_2_,0)]	[(b_1_,0),(b_1_,0)]	[(a_0_,0),(a_0_,0)]
FM_2_-CF_8_	[(c_0_,0),(c_0_,0)]	[(a_2_,0),(a_2_,0)]	[(a_2_,0),(a_2_,0)]	[(b_2_,0),(b_2_,0)]	[(a_0_,0),(a_0_,0)]
FM_3_-CF_9_	[(c_0_,0),(c_0_,0)]	[(a_1_,0),(a_1_,0)]	[(a_2_,0),(a_2_,0)]	[(b_2_,0),(b_2_,0)]	[(a_0_,0),(a_0_,0)]
FM_3_-CF_10_	[(c_1_,0),(c_1_,0)]	[(a_1_,0),(a_1_,0)]	[(a_2_,0),(a_2_,0)]	[(b_3_,0),(b_3_,0)]	[(a_0_,0),(a_0_,0)]
FM_4_-CF_3_	[(c_0_,0),(c_0_,0)]	[(a_0_,0),(a_0_,0)]	[(a_4_,0),(a_4_,0)]	[(b_1_,0),(b_1_,0)]	[(a_0_,0),(a_0_,0)]
FM_4_-CF_11_	[(c_4_,0),(c_4_,0)]	[(a_0_,0),(a_0_,0)]	[(a_4_,0),(a_4_,0)]	[(b_1_,0),(b_1_,0)]	[(a_0_,0),(a_0_,0)]
FM_4_-CF_12_	[(c_4_,0),(c_4_,0)]	[(a_0_,0),(a_0_,0)]	[(a_2_,0),(a_2_,0)]	[(b_1_,0),(b_1_,0)]	[(a_0_,0),(a_0_,0)]
FM_5_-CF_13_	[(c_0_,0),(c_0_,0)]	[(a_0_,0),(a_0_,0)]	[(a_4_,0),(a_4_,0)]	[(b_2_,0),(b_2_,0)]	[(a_0_,0),(a_0_,0)]
FM_5_-CF_14_	[(c_0_,0),(c_0_,0)]	[(a_3_,0),(a_3_,0)]	[(a_4_,0),(a_4_,0)]	[(b_3_,0),(b_3_,0)]	[(a_1_,0),(a_1_,0)]
FM_5_-CF_15_	[(c_4_,0),(c_4_,0)]	[(a_0_,0),(a_0_,0)]	[(a_2_,0),(a_2_,0)]	[(b_3_,0),(b_3_,0)]	[(a_1_,0),(a_1_,0)]
FM_5_-CF_16_	[(c_4_,0),(c_4_,0)]	[(a_0_,0),(a_0_,0)]	[(a_2_,0),(a_2_,0)]	[(b_3_,0),(b_3_,0)]	[(a_1_,0),(a_1_,0)]
FM_5_-CF_17_	[(c_2_,0),(c_2_,0)]	[(a_0_,0),(a_0_,0)]	[(a_4_,0),(a_4_,0)]	[(b_2_,0),(b_2_,0)]	[(a_0_,0),(a_0_,0)]
FM_5_-CF_18_	[(c_2_,0),(c_2_,0)]	[(a_0_,0),(a_0_,0)]	[(a_4_,0),(a_4_,0)]	[(b_3_,0),(b_3_,0)]	[(a_0_,0),(a_0_,0)]
FM_5_-CF_19_	[(c_2_,0),(c_2_,0)]	[(a_0_,0),(a_0_,0)]	[(a_4_,0),(a_4_,0)]	[(b_3_,0),(b_3_,0)]	[(a_0_,0),(a_0_,0)]
FM_5_-CF_20_		[(a_2_,0),(a_2_,0)]	[(a_2_,0),(a_2_,0)]	[(b_2_,0),(b_2_,0)]	[(a_0_,0),(a_0_,0)]
FM_5_-CF_21_	[(c_4_,0),(c_4_,0)]	[(a_2_,0),(a_2_,0)]	[(a_2_,0),(a_2_,0)]	[(b_2_,0),(b_2_,0)]	[(a_0_,0),(a_0_,0)]
FM_5_-CF_22_	[(c_4_,0),(c_4_,0)]	[(a_2_,0),(a_2_,0)]	[(a_2_,0),(a_2_,0)]	[(b_2_,0),(b_2_,0)]	[(a_0_,0),(a_0_,0)]
FM_5_-CF_23_	[(c_0_,0),(c_0_,0)]	[(a_2_,0),(a_2_,0)]	[(a_4_,0),(a_4_,0)]	[(b_2_,0),(b_2_,0)]	[(a_0_,0),(a_0_,0)]
FM_6_-CF_4_	[(c_1_,0),(c_1_,0)]	[(a_2_,0),(a_2_,0)]	[(a_4_,0),(a_4_,0)]	[(b_2_,0),(b_2_,0)]	[(a_0_,0),(a_0_,0)]
FM_6_-CF_24_		[(a_1_,0),(a_1_,0)]	[(a_4_,0),(a_4_,0)]	[(b_2_,0),(b_2_,0)]	[(a_0_,0),(a_0_,0)]
FM_6_-CF_25_	[(c_1_,0),(c_1_,0)]	[(a_1_,0),(a_1_,0)]	[(a_3_,0),(a_3_,0)]	[(b_3_,0),(b_3_,0)]	[(a_0_,0),(a_0_,0)]
FM_7_-CF_16_	[(c_4_,0),(c_4_,0)]	[(a_1_,0),(a_1_,0)]	[(a_2_,0),(a_2_,0)]	[(b_3_,0),(b_3_,0)]	[(a_1_,0),(a_1_,0)]
FM_7_-CF_22_	[(c_1_,0),(c_1_,0)]	[(a_0_,0),(a_0_,0)]	[(a_2_,0),(a_2_,0)]	[(b_2_,0),(b_2_,0)]	[(a_0_,0),(a_0_,0)]
FM_7_-CF_26_	[(c_4_,0),(c_4_,0)]	[(a_2_,0),(a_2_,0)]	[(a_2_,0),(a_2_,0)]	[(b_3_,0),(b_3_,0)]	[(a_0_,0),(a_0_,0)]
FM_7_-CF_27_	[(c_2_,0),(c_2_,0)]	[(a_2_,0),(a_2_,0)]	[(a_2_,0),(a_2_,0)]	[(b_2_,0),(b_2_,0)]	[(a_0_,0),(a_0_,0)]

**Table 5 healthcare-10-02283-t005:** Interval 2-tuple linguistic variables for Detection (D).

	DM_1_	DM_2_	DM_3_	DM_4_	DM_5_
FM_1_-CF_1_	[(c_0_,0),(c_0_,0)]	[(a_4_,0),(a_4_,0)]	[(a_4_,0),(a_4_,0)]	[(b_2_,0),(b_2_,0)]	[(a_0_,0),(a_0_,0)]
FM_1_-CF_2_	[(c_0_,0),(c_0_,0)]	[(a_0_,0),(a_0_,0)]	[(a_4_,0),(a_4_,0)]	[(b_1_,0),(b_1_,0)]	[(a_0_,0),(a_0_,0)]
FM_1_-CF_3_	[(c_1_,0),(c_1_,0)]	[(a_1_,0),(a_1_,0)]	[(a_4_,0),(a_4_,0)]	[(b_2_,0),(b_2_,0)]	[(a_0_,0),(a_0_,0)]
FM_1_-CF_4_	[(c_0_,0),(c_0_,0)]	[(a_1_,0),(a_1_,0)]	[(a_4_,0),(a_4_,0)]	[(b_1_,0),(b_1_,0)]	[(a_1_,0),(a_1_,0)]
FM_1_-CF_5_	[(c_0_,0),(c_0_,0)]	[(a_0_,0),(a_0_,0)]	[(a_4_,0),(a_4_,0)]	[(b_3_,0),(b_3_,0)]	[(a_0_,0),(a_0_,0)]
FM_1_-CF_6_	[(c_7_,0),(c_7_,0)]	[(a_2_,0),(a_2_,0)]	[(a_4_,0),(a_4_,0)]	[(b_1_,0),(b_1_,0)]	[(a_0_,0),(a_0_,0)]
FM_1_-CF_7_	[(c_6_,0),(c_6_,0)]	[(a_1_,0),(a_1_,0)]	[(a_4_,0),(a_4_,0)]	[(b_3_,0),(b_3_,0)]	[(a_0_,0),(a_0_,0)]
FM_2_-CF_5_	[(c_0_,0),(c_0_,0)]	[(a_0_,0),(a_0_,0)]	[(a_4_,0),(a_4_,0)]	[(b_3_,0),(b_3_,0)]	[(a_0_,0),(a_0_,0)]
FM_2_-CF_8_	[(c_0_,0),(c_0_,0)]	[(a_0_,0),(a_0_,0)]	[(a_4_,0),(a_4_,0)]	[(b_3_,0),(b_3_,0)]	[(a_0_,0),(a_0_,0)]
FM_3_-CF_9_	[(c_6_,0),(c_6_,0)]	[(a_0_,0),(a_0_,0)]	[(a_4_,0),(a_4_,0)]	[(b_0_,0),(b_0_,0)]	[(a_0_,0),(a_0_,0)]
FM_3_-CF_10_	[(c_6_,0),(c_6_,0)]	[(a_0_,0),(a_0_,0)]	[(a_4_,0),(a_4_,0)]	[(b_1_,0),(b_1_,0)]	[(a_0_,0),(a_0_,0)]
FM_4_-CF_3_	[(c_1_,0),(c_1_,0)]	[(a_2_,0),(a_2_,0)]	[(a_4_,0),(a_4_,0)]	[(b_1_,0),(b_1_,0)]	[(a_0_,0),(a_0_,0)]
FM_4_-CF_11_	[(c_8_,0),(c_8_,0)]	[(a_0_,0),(a_0_,0)]	[(a_4_,0),(a_4_,0)]	[(b_4_,0),(b_4_,0)]	[(a_0_,0),(a_0_,0)]
FM_4_-CF_12_	[(c_4_,0),(c_4_,0)]	[(a_2_,0),(a_2_,0)]	[(a_4_,0),(a_4_,0)]	[(b_1_,0),(b_1_,0)]	[(a_0_,0),(a_0_,0)]
FM_5_-CF_13_	[(c_0_,0),(c_0_,0)]	[(a_0_,0),(a_0_,0)]	[(a_4_,0),(a_4_,0)]	[(b_4_,0),(b_4_,0)]	[(a_0_,0),(a_0_,0)]
FM_5_-CF_14_	[(c_0_,0),(c_0_,0)]	[(a_4_,0),(a_4_,0)]	[(a_4_,0),(a_4_,0)]	[(b_1_,0),(b_1_,0)]	[(a_0_,0),(a_0_,0)]
FM_5_-CF_15_	[(c_6_,0),(c_6_,0)]	[(a_1_,0),(a_1_,0)]	[(a_4_,0),(a_4_,0)]	[(b_1_,0),(b_1_,0)]	[(a_0_,0),(a_0_,0)]
FM_5_-CF_16_	[(c_4_,0),(c_4_,0)]	[(a_0_,0),(a_0_,0)]	[(a_4_,0),(a_4_,0)]	[(b_1_,0),(b_1_,0)]	[(a_0_,0),(a_0_,0)]
FM_5_-CF_17_	[(c_6_,0),(c_6_,0)]	[(a_0_,0),(a_0_,0)]	[(a_4_,0),(a_4_,0)]	[(b_1_,0),(b_1_,0)]	[(a_0_,0),(a_0_,0)]
FM_5_-CF_18_	[(c_6_,0),(c_6_,0)]	[(a_0_,0),(a_0_,0)]	[(a_4_,0),(a_4_,0)]	[(b_1_,0),(b_1_,0)]	[(a_0_,0),(a_0_,0)]
FM_5_-CF_19_	[(c_6_,0),(c_6_,0)]	[(a_0_,0),(a_0_,0)]	[(a_4_,0),(a_4_,0)]	[(b_1_,0),(b_1_,0)]	[(a_1_,0),(a_1_,0)]
FM_5_-CF_20_		[(a_4_,0),(a_4_,0)]	[(a_4_,0),(a_4_,0)]	[(b_3_,0),(b_3_,0)]	[(a_1_,0),(a_1_,0)]
FM_5_-CF_21_	[(c_6_,0),(c_6_,0)]	[(a_4_,0),(a_4_,0)]	[(a_4_,0),(a_4_,0)]	[(b_4_,0),(b_4_,0)]	[(a_0_,0),(a_0_,0)]
FM_5_-CF_22_	[(c_4_,0),(c_4_,0)]	[(a_4_,0),(a_4_,0)]	[(a_4_,0),(a_4_,0)]	[(b_4_,0),(b_4_,0)]	[(a_0_,0),(a_0_,0)]
FM_5_-CF_23_	[(c_2_,0),(c_2_,0)]	[(a_1_,0),(a_1_,0)]	[(a_4_,0),(a_4_,0)]	[(b_1_,0),(b_1_,0)]	[(a_0_,0),(a_0_,0)]
FM_6_-CF_4_	[(c_8_,0),(c_8_,0)]	[(a_0_,0),(a_0_,0)]	[(a_4_,0),(a_4_,0)]	[(b_1_,0),(b_1_,0)]	[(a_0_,0),(a_0_,0)]
FM_6_-CF_24_		[(a_1_,0),(a_1_,0)]	[(a_4_,0),(a_4_,0)]	[(b_1_,0),(b_1_,0)]	[(a_0_,0),(a_0_,0)]
FM_6_-CF_25_	[(c_1_,0),(c_1_,0)]	[(a_1_,0),(a_1_,0)]	[(a_4_,0),(a_4_,0)]	[(b_1_,0),(b_1_,0)]	[(a_0_,0),(a_0_,0)]
FM_7_-CF_16_	[(c_2_,0),(c_2_,0)]	[(a_2_,0),(a_2_,0)]	[(a_4_,0),(a_4_,0)]	[(b_4_,0),(b_4_,0)]	[(a_1_,0),(a_1_,0)]
FM_7_-CF_22_	[(c_7_,0),(c_7_,0)]	[(a_1_,0),(a_1_,0)]	[(a_4_,0),(a_4_,0)]	[(b_5_,0),(b_5_,0)]	[(a_1_,0),(a_1_,0)]
FM_7_-CF_26_	[(c_4_,0),(c_4_,0)]	[(a_2_,0),(a_2_,0)]	[(a_4_,0),(a_4_,0)]	[(b_4_,0),(b_4_,0)]	[(a_1_,0),(a_1_,0)]
FM_7_-CF_27_	[(c_4_,0),(c_4_,0)]	[(a_1_,0),(a_1_,0)]	[(a_4_,0),(a_4_,0)]	[(b_1_,0),(b_1_,0)]	[(a_1_,0),(a_1_,0)]

**Table 6 healthcare-10-02283-t006:** Interval 2-tuple linguistic variables for the importance of risk factors.

	DM_1_	DM_2_	DM_3_	DM_4_	DM_5_
S	(d_1_,0)	(d_4_,0)	(d_3_,0)	(d_4_,0)	(d_2_,0)
O	(d_2_,0)	(d_4_,0)	(d_3_,0)	(d_3_,0)	(d_1_,0)
D	(d_3_,0)	(d_4_,0)	(d_3_,0)	(d_3_,0)	(d_3_,0)

**Table 7 healthcare-10-02283-t007:** Interval 2-tuple linguistic decision matrix $\tilde{R}$.

	S	O	D
FM_1_-CF_1_	Δ[0.467,0.467]	Δ[0.542,0.542]	Δ[0.467,0.467]
FM_1_-CF_2_	Δ[0.350,0.350]	Δ[0.342,0.342]	Δ[0.233,0.233]
FM_1_-CF_3_	Δ[0.200,0.200]	Δ[0.417,0.417]	Δ[0.342,0.342]
FM_1_-CF_4_	Δ[0.250,0.250]	Δ[0.533,0.533]	Δ[0.333,0.333]
FM_1_-CF_5_	Δ[0.250,0.250]	Δ[0.283,0.283]	Δ[0.300,0.300]
FM_1_-CF_6_	Δ[0.458,0.458]	Δ[0.358,0.358]	Δ[0.508,0.508]
FM_1_-CF_7_	Δ[0.458,0.458]	Δ[0.392,0.392]	Δ[0.500,0.500]
FM_2_-CF_5_	Δ[0.200,0.200]	Δ[0.283,0.283]	Δ[0.300,0.300]
FM_2_-CF_8_	Δ[0.283,0.283]	Δ[0.267,0.267]	Δ[0.300,0.300]
FM_3_-CF_9_	Δ[0.417,0.417]	Δ[0.217,0.217]	Δ[0.350,0.350]
FM_3_-CF_10_	Δ[0.400,0.400]	Δ[0.275,0.275]	Δ[0.383,0.383]
FM_4_-CF_3_	Δ[0.467,0.467]	Δ[0.233,0.233]	Δ[0.358,0.358]
FM_4_-CF_11_	Δ[0.400,0.400]	Δ[0.333,0.333]	Δ[0.533,0.533]
FM_4_-CF_12_	Δ[0.275,0.275]	Δ[0.233,0.233]	Δ[0.433,0.433]
FM_5_-CF_13_	Δ[0.467,0.467]	Δ[0.267,0.267]	Δ[0.333,0.333]
FM_5_-CF_14_	Δ[0.600,0.600]	Δ[0.500,0.500]	Δ[0.433,0.433]
FM_5_-CF_15_	Δ[0.400,0.400]	Δ[0.350,0.350]	Δ[0.433,0.433]
FM_5_-CF_16_	Δ[0.450,0.450]	Δ[0.350,0.350]	Δ[0.333,0.333]
FM_5_-CF_17_	Δ[0.583,0.583]	Δ[0.317,0.317]	Δ[0.383,0.383]
FM_5_-CF_18_	Δ[0.583,0.583]	Δ[0.350,0.350]	Δ[0.383,0.383]
FM_5_-CF_19_	Δ[0.483,0.483]	Δ[0.350,0.350]	Δ[0.433,0.433]
FM_5_-CF_20_	Δ[0.400,0.400]	Δ[0.300,0.300]	Δ[0.150,0.150]
FM_5_-CF_21_	Δ[0.400,0.400]	Δ[0.267,0.267]	Δ[0.533,0.533]
FM_5_-CF_22_	Δ[0.550,0.550]	Δ[0.367,0.367]	Δ[0.633,0.633]
FM_5_-CF_23_	Δ[0.433,0.433]	Δ[0.367,0.367]	Δ[0.333,0.333]
FM_6_-CF_4_	Δ[0.658,0.658]	Δ[0.392,0.392]	Δ[0.433,0.433]
FM_6_-CF_24_	Δ[0.450,0.450]	Δ[0.317,0.317]	Δ[0.283,0.283]
FM_6_-CF_25_	Δ[0.458,0.458]	Δ[0.325,0.325]	Δ[0.308,0.308]
FM_7_-CF_16_	Δ[0.550,0.550]	Δ[0.400,0.400]	Δ[0.533,0.533]
FM_7_-CF_22_	Δ[0.542,0.542]	Δ[0.192,0.192]	Δ[0.642,0.642]
FM_7_-CF_26_	Δ[0.467,0.467]	Δ[0.400,0.400]	Δ[0.583,0.583]
FM_7_-CF_27_	Δ[0.317,0.317]	Δ[0.317,0.317]	Δ[0.433,0.433]
Weight	0.700	0.650	0.800
Normalized Weight	0.326	0.302	0.372

**Table 8 healthcare-10-02283-t008:** The distance matrix $D_{0}$ for pairs of FMs and CFs.

Pairs	S	O	D
FM_1_-CF_1_	Δ(3.533)	Δ(3.458)	Δ(3.533)
FM_1_-CF_2_	Δ(3.650)	Δ(3.658	Δ(3.767)
FM_1_-CF_3_	Δ(3.800)	Δ(3.583)	Δ(3.658)
FM_1_-CF_4_	Δ(3.750)	Δ(3.467)	Δ(3.667)
FM_1_-CF_5_	Δ(3.750)	Δ(3.717)	Δ(3.700)
FM_1_-CF_6_	Δ(3.542)	Δ(3.642)	Δ(3.492)
FM_1_-CF_7_	Δ(3.542)	Δ(3.608)	Δ(3.500)
FM_2_-CF_5_	Δ(3.800)	Δ(3.717)	Δ(3.700)
FM_2_-CF_8_	Δ(3.717)	Δ(3.733)	Δ(3.700)
FM_3_-CF_9_	Δ(3.583)	Δ(3.783)	Δ(3.650)
FM_3_-CF_10_	Δ(3.600)	Δ(3.725)	Δ(3.617)
FM_4_-CF_3_	Δ(3.533)	Δ(3.767)	Δ(3.642)
FM_4_-CF_11_	Δ(3.600)	Δ(3.667)	Δ(3.467)
FM_4_-CF_12_	Δ(3.725)	Δ(3.767)	Δ(3.567)
FM_5_-CF_13_	Δ(3.533)	Δ(3.733)	Δ(3.667)
FM_5_-CF_14_	Δ(3.400)	Δ(3.500)	Δ(3.567)
FM_5_-CF_15_	Δ(3.600)	Δ(3.650)	Δ(3.567)
FM_5_-CF_16_	Δ(3.550)	Δ(3.650)	Δ(3.667)
FM_5_-CF_17_	Δ(3.417)	Δ(3.683)	Δ(3.617)
FM_5_-CF_18_	Δ(3.417)	Δ(3.650)	Δ(3.617)
FM_5_-CF_19_	Δ(3.517)	Δ(3.650)	Δ(3.567)
FM_5_-CF_20_	Δ(3.600)	Δ(3.700)	Δ(3.850)
FM_5_-CF_21_	Δ(3.600)	Δ(3.733)	Δ(3.467)
FM_5_-CF_22_	Δ(3.450)	Δ(3.633)	Δ(3.367)
FM_5_-CF_23_	Δ(3.567)	Δ(3.633)	Δ(3.667)
FM_6_-CF_4_	Δ(3.342)	Δ(3.608)	Δ(3.567)
FM_6_-CF_24_	Δ(3.550)	Δ(3.683)	Δ(3.717)
FM_6_-CF_25_	Δ(3.542)	Δ(3.675)	Δ(3.692)
FM_7_-CF_16_	Δ(3.450)	Δ(3.600)	Δ(3.467)
FM_7_-CF_22_	Δ(3.458)	Δ(3.808)	Δ(3.358)
FM_7_-CF_26_	Δ(3.533)	Δ(3.600)	Δ(3.417)
FM_7_-CF_27_	Δ(3.683)	Δ(3.683)	Δ(3.567)

**Table 9 healthcare-10-02283-t009:** The grey relational coefficient for pairs of FMs and CFs.

Pairs	S	O	D
FM1-CF1	Δ(0.965)	Δ(0.978)	Δ(0.965)
FM1-CF2	Δ(0.945)	Δ(0.943)	Δ(0.925)
FM1-CF3	Δ(0.920)	Δ(0.956)	Δ(0.943)
FM1-CF4	Δ(0.928)	Δ(0.977)	Δ(0.942)
FM1-CF5	Δ(0.928)	Δ(0.934)	Δ(0.936)
FM1-CF6	Δ(0.963)	Δ(0.946)	Δ(0.972)
FM1-CF7	Δ(0.963)	Δ(0.952)	Δ(0.971)
FM2-CF5	Δ(0.920)	Δ(0.934)	Δ(0.936)
FM2-CF8	Δ(0.934)	Δ(0.931)	Δ(0.936)
FM3-CF9	Δ(0.956)	Δ(0.923)	Δ(0.945)
FM3-CF10	Δ(0.953)	Δ(0.932)	Δ(0.950)
FM4-CF3	Δ(0.965)	Δ(0.925)	Δ(0.946)
FM4-CF11	Δ(0.953)	Δ(0.942)	Δ(0.977)
FM4-CF12	Δ(0.932)	Δ(0.925)	Δ(0.959)
FM5-CF13	Δ(0.965)	Δ(0.931)	Δ(0.942)
FM5-CF14	Δ(0.989)	Δ(0.971)	Δ(0.959)
FM5-CF15	Δ(0.953)	Δ(0.945)	Δ(0.959)
FM5-CF16	Δ(0.962)	Δ(0.945)	Δ(0.942)
FM5-CF17	Δ(0.986)	Δ(0.939)	Δ(0.950)
FM5-CF18	Δ(0.986)	Δ(0.945)	Δ(0.950)
FM5-CF19	Δ(0.968)	Δ(0.945)	Δ(0.959)
FM5-CF20	Δ(0.953)	Δ(0.936)	Δ(0.912)
FM5-CF21	Δ(0.953)	Δ(0.931)	Δ(0.977)
FM5-CF22	Δ(0.980)	Δ(0.948)	Δ(0.995)
FM5-CF23	Δ(0.959)	Δ(0.948)	Δ(0.942)
FM6-CF4	Δ(1.000)	Δ(0.952)	Δ(0.959)
FM6-CF24	Δ(0.962)	Δ(0.939)	Δ(0.934)
FM6-CF25	Δ(0.963)	Δ(0.941)	Δ(0.938)
FM7-CF16	Δ(0.980)	Δ(0.953)	Δ(0.977)
FM7-CF22	Δ(0.978)	Δ(0.919)	Δ(0.997)
FM7-CF26	Δ(0.965)	Δ(0.953)	Δ(0.986)
FM7-CF27	Δ(0.939)	Δ(0.939)	Δ(0.959)

**Table 10 healthcare-10-02283-t010:** The grey relational degree of FM–CF pairs.

Pairs	Grey Relational Degree	2-Tuple
FM1-CF1	Δ(0.969)	(a4,-0.030)
FM1-CF2	Δ(0.937)	(a4,-0.063)
FM1-CF3	Δ(0.940)	(a4,-0.060)
FM1-CF4	Δ(0.948)	(a4,-0.052)
FM1-CF5	Δ(0.933)	(a4,-0.067)
FM1-CF6	Δ(0.962)	(a4,-0.038)
FM1-CF7	Δ(0.963)	(a4,-0.037)
FM2-CF5	Δ(0.930)	(a4,-0.069)
FM2-CF8	Δ(0.934)	(a4,-0.066)
FM3-CF9	Δ(0.942)	(a4,-0.058)
FM3-CF10	Δ(0.946)	(a4,-0.054)
FM4-CF3	Δ(0.946)	(a4,-0.053)
FM4-CF11	Δ(0.959)	(a4,-0.041)
FM4-CF12	Δ(0.940)	(a4,-0.060)
FM5-CF13	Δ(0.946)	(a4,-0.054)
FM5-CF14	Δ(0.972)	(a4,-0.027)
FM5-CF15	Δ(0.953)	(a4,-0.047)
FM5-CF16	Δ(0.949)	(a4,-0.051)
FM5-CF17	Δ(0.959)	(a4,-0.041)
FM5-CF18	Δ(0.960)	(a4,-0.040)
FM5-CF19	Δ(0.958)	(a4,-0.042)
FM5-CF20	Δ(0.933)	(a4,-0.067)
FM5-CF21	Δ(0.955)	(a4,-0.045)
FM5-CF22	Δ(0.976)	(a4,-0.024)
FM5-CF23	Δ(0.949)	(a4,-0.055)
FM6-CF4	Δ(0.970)	(a4,-0.030)
FM6-CF24	Δ(0.945)	(a4,-0.055)
FM6-CF25	Δ(0.947)	(a4,-0.053)
FM7-CF16	Δ(0.971)	(a4,-0.029)
FM7-CF22	Δ(0.967)	(a4,-0.033)
FM7-CF26	Δ(0.969)	(a4,-0.031)
FM7-CF27	Δ(0.947)	(a4,-0.053)

**Table 11 healthcare-10-02283-t011:** The initial relation matrix *Y*.

H	CF_1_	CF_2_	…	CF_26_	CF_27_	FM_1_	FM_2_	…	FM_6_	FM_7_
**CF_1_**						0.937				
**CF_2_**						0.906				
**…**										
**CF_26_**										0.937
**CF_27_**										0.945
**FM_1_**										
**FM_2_**										
**…**										
**FM_6_**										
**FM_7_**										

**Table 12 healthcare-10-02283-t012:** The direct and indirect relation matrix *T* among FMs and CFs.

H	CF_1_	CF_2_	…	CF_26_	CF_27_	FM_1_	FM_2_	…	FM_6_	FM_7_
**CF_1_**						0.499				
**CF_2_**						0.482				
**…**										
**CF_26_**										0.499
**CF_27_**										0.503
**FM_1_**										
**FM_2_**										
**…**										
**FM_6_**										
**FM_7_**										

**Table 13 healthcare-10-02283-t013:** Values of R for CFs.

CFs	R	CFs	R
CF_1_	0.499	CF_15_	0.490
CF_2_	0.482	CF_16_	**0.988**
CF_3_	**0.970**	CF_17_	0.493
CF_4_	**0.987**	CF_18_	0.494
CF_5_	**0.959**	CF_19_	0.493
CF_6_	0.495	CF_20_	0.480
CF_7_	0.495	CF_21_	0.492
CF_8_	0.481	CF_22_	**1.000**
CF_9_	0.485	CF_23_	0.488
CF_10_	0.487	CF_24_	0.486
CF_11_	0.493	CF_25_	0.487
CF_12_	0.484	CF_26_	0.499
CF_13_	0.487	CF_27_	0.503
CF_14_	0.500		

**Table 14 healthcare-10-02283-t014:** Values of C for FMs.

FMs	C	FMs	C
FM_1_	**3.423**	FM_5_	**5.409**
FM_2_	0.959	FM_6_	1.473
FM_3_	0.972	FM_7_	**1.999**
FM_4_	1.464		

**Table 15 healthcare-10-02283-t015:** Priority of CFs relevant to the top three FMs.

No.	CF	FM	R + C	Priority	No.	CF	FM	R + C	Priority
1	CF_22_	FM_5_	1.000	1	12	CF_7_	FM_1_	0.495	10
2	CF_22_	FM_7_	1.000	1	13	CF_18_	FM_5_	0.494	11
3	CF_16_	FM_5_	0.988	2	14	CF_17_	FM_5_	0.493	13
4	CF_16_	FM_7_	0.988	2	15	CF_19_	FM_5_	0.493	14
5	CF_4_	FM_1_	0.987	3	16	CF_21_	FM_5_	0.492	15
6	CF_3_	FM_1_	0.970	4	17	CF_15_	FM_5_	0.490	16
7	CF_5_	FM_1_	0.959	5	18	CF_23_	FM_5_	0.488	17
8	CF_14_	FM_5_	0.500	6	19	CF_27_	FM_7_	0.487	21
9	CF_1_	FM_1_	0.499	8	20	CF_2_	FM_1_	0.482	25
10	CF_26_	FM_7_	0.499	8	21	CF_20_	FM_5_	0.48	27
11	CF_6_	FM_1_	0.495	9	22	CF_13_	FM_5_	0.487	29

**Table 16 healthcare-10-02283-t016:** Comparison regarding the priorities of FMs.

	GD-FMEA	G-FMEA	CFFS-VIKOR-A	CFFS-VIRO-C
FM1	2	2	2	2
FM2	7	7	7	7
FM3	6	6	6	6
FM4	5	5	4	3
FM5	1	1	1	1
FM6	4	4	3	4
FM7	3	3	5	5

**Table 17 healthcare-10-02283-t017:** Comparison regarding the priority of CFs related to FM_1_, FM_4_, FM_5_, FM_6_ and FM_7_, which are ranked as the top three by at least one of the compared approaches.

		CFFS-VIKOR-A	CFFS-VIKOR-C	G-FMEA	GD-FMEA
FM_1_	CF_1_	1	6	6	8
FM_1_	CF_2_	23	24	24	25
FM_1_	CF_3_	12	12	19	4
FM_1_	CF_4_	2	2	4	3
FM_1_	CF_5_	26	27	27	5
FM_1_	CF_6_	7	4	8	9
FM_1_	CF_7_	19	17	7	10
FM_4_	CF_3_	12	12	19	4
FM_4_	CF_11_	5	1	10	12
FM_4_	CF_12_	18	20	23	24
FM_5_	CF_13_	10	10	18	29
FM_5_	CF_14_	3	3	2	6
FM_5_	CF_15_	27	22	14	16
FM_5_	CF_16_	15	13	3	2
FM_5_	CF_17_	9	9	11	13
FM_5_	CF_18_	4	7	9	11
FM_5_	CF_19_	17	16	12	14
FM_5_	CF_20_	11	18	26	27
FM_5_	CF_21_	13	5	13	15
FM_5_	CF_22_	8	8	1	1
FM_5_	CF_23_	6	14	15	17
FM_6_	CF_4_	2	2	4	3
FM_6_	CF_24_	16	15	21	22
FM_6_	CF_25_	14	11	16	20
FM_7_	CF_16_	15	13	3	2
FM_7_	CF_22_	8	8	1	1
FM_7_	CF_26_	20	19	5	8
FM_7_	CF_27_	25	26	17	21

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
