# Peer review of "Identifying Root Causes of Important Service Failures across Medical Examination Processes with Integration of 4M1E, ITLV, GRA, DEMATEL and FMEA"

_healthcare, 2022, doi:10.3390/healthcare10112283_

Round 1

Reviewer 1 Report

-Reference citation is not is order of appearance in text (e.g. 5 is followed by 13)

Abbreviation like 4M1E, RPN, ITLV should be explained for the non-professional readers

The obtained results comparison with other similar studies should be compared

The occurrence of the mentioned FMs and CFs it could be an objective parameter (e.g. there are software problems about e.g. 4 times/month), the distance between c4 and a0 suggest that it is about an “impression” about the occurrence of the mentioned problem.

In the Table 2 at CF24 all answers are given in C mode….

The conclusions/discussions are superficial, it can be stated, without making the study, that “high-quality staff training, standardization of operation manual and improvement of machine maintenance are quite essential”

There are some factors, without them the others (even all of them) can work greatly, the result (a satisfied patient) will not be good

Author Response

Dear Editors and Reviewers,

We would like to thank you very much for our patience and we appreciate the editors and reviewer for your valuable comments and suggestions on our manuscript entitled “Improving quality of services relevant with medical examination through DEMATEL-based Failure Mode and Effect Analysis”. We have studied the comments and suggestions carefully and have made the revisions which hopefully meet with expectations.

The main corrections in the revised manuscript are colored in yellow and point-to-point responses to the reviewers’ comments, which are in red, are as follows:

  1. Reference citation is not in order of appearance in text (e.g. 5 is followed by 13)

Response:

  We have adjusted the citation according to the order of appearance in text.

  1. Abbreviation like 4M1E, RPN, ITLV should be explained for the non-professional readers.

Response:

 Full names of those abbreviations have been added in the revised version.

  1. The obtained results comparison with other similar studies should be compared

Response:

The results obtained by the proposed GD-FMEA (the combination of GRA, DEMATEL and FMEA) have been compared with those obtained by G-FMEA (the combination of GRA and FMEA) and CFFS-VIKOR (the evaluation approach proposed in [13]) respectively. The details of the comparison results are shown in Table 13, and the discussion about the comparison results can be observed in section 4.

  1. The occurrence of the mentioned FMs and CFs it could be an objective parameter (e.g. there are software problems about e.g. 4 times/month), the distance between c4 and a0 suggest that it is about an “impression” about the occurrence of the mentioned problem. In the Table 2 at CF24 all answers are given in C mode…

Response:

Thank you very much for your comments. Firstly, we would like to explain that since the information system of the targeted hospital is not capable of recording automatically the occurrence of FMs, we take the evaluation of professionals who have been working in this department for quite a long period as a valuable substitute.

 After carefully checking Table 2, we observed that all answers about CF27, instead of CF24,were given in C mode, which was incorrect. And we have corrected the Table 2 and other relevant tables.

  1. The conclusions/discussions are superficial, it can be stated, without making the study, that “high-quality staff training, standardization of operation manual and improvement of machine maintenance are quite essential”

Response:

The conclusions have been improved to illustrate the observation based on the experimental results obtained in this study, and details are given in section 5.

Reviewer 2 Report

This is a solid research paper on a quality control model. It provides a detailed description of the model. The model estimation results support the conclusion.

However, as a research paper publishable in Healthcare, we need to consider feasibility in public health practice. For example, although some measures are better than others, there are still issues with cost-benefit concerns. If the authors address these issues a little bit, e.g., implications to practice, conditions/resources, and limitations, e.g., in the discussion section or conclusion section.  

Author Response

Dear Editors and Reviewers,

We would like to thank you very much for our patience and we appreciate the editors and reviewer for your valuable comments and suggestions on our manuscript entitled “Improving quality of services relevant with medical examination through DEMATEL-based Failure Mode and Effect Analysis”. We have studied the comments and suggestions carefully and have made the revisions which hopefully meet with expectations.

The main corrections in the revised manuscript are colored in yellow and point-to-point responses to the reviewers’ comments, which are in red, are as follows:

  1. The abstract is not convincing and is disorganized, it should be refined to precisely illustrate what authors have done in this paper within 200 words. The abstract must be a concise yet comprehensive reflection of what is in your paper. Remember that reader want to know:  1-what is the problem. 2- why the problem is relevant 3- wants an overview of your approach. 4-need to know the results.

Response:

   The abstract has been carefully modified according to your valuable comments and we hope it can meet with expectations.  

  1. Manuscript needs a good introduction, the introduction section of the manuscript is weak, authors are advised to improvise the introduction section. In the Introduction part, the new features of the proposed method and the main advantages of the results over others should be clearly described. An introduction should clearly highlight the motivation, problem statement, the objective of the paper, gap in the existing research and the novelty of the conducted research.

Response:

 The part of introduction in the revised version have been improved by (1) citing more valuable references, (2) highlighting the motivation and the purpose of this study, (3) rationalizing the methodology applied in this study, and (4) introducing novelty of this study by give details about the techniques applied to enhance the traditional model.  

  1. This application topic has not received much attention in the literature. However, the study, literature review and presentation require substantial improvement in several respects.

Response:

  The literature review, as a part of the introduction, has been improved in the revised version.

  1. When I checked the results, I noticed that there were mistakes, please recheck.

Response

  After carefully checking results, we observed that all answers about CF27 in Table 2 were given in C mode, which was incorrect. And we have corrected the corresponding results.

  1. I suggest extending the conclusions section to focus on the results you get, the method you propose, and their significance.

Response:

  The part of conclusions has been improved in the revised version, supporting with more experimental results.

  1. The article need to be revised with more experimentation by comparison relevant approach and algorithms.

Response:

 One more benchmark proposed in one of the most recent work is compared with our proposed method to verify the quality of your assessment approach. Details of the experimental results are given in Table 13 and discussions are improved in section 4.

  1. Expand literature review and cite most related work:

Tuple Linguistic Fermatean Fuzzy Decision-Making Method Based on COCOSO with CRITIC for Drip Irrigation System Analysis. Journal of Computational and Cognitive Engineering. https://doi.org/10.47852/bonviewJCCE2202356

Extended CODAS method for multi-attribute group decision-making based on 2-tuple linguistic Fermatean fuzzy Hamacher aggregation operators. Granul. Comput. (2022). https://doi.org/10.1007/s41066-022-00332-3

Assessment of Hydropower Plants in Pakistan: Muirhead Mean-Based 2-Tuple Linguistic T-spherical Fuzzy Model Combining SWARA with COPRAS. Arab J Sci Eng (2022). https://doi.org/10.1007/s13369-022-07081-0

Extending COPRAS Method with Linguistic Fermatean Fuzzy Sets and Hamy Mean Operators, Journal of Mathematics, Volume 2022 | Article ID 8239263 | https://doi.org/10.1155/2022/8239263.

Response:

 Thank you very much for recommending these valuable references and they have been cited in the revised version.

Reviewer 3 Report

This is a nice article. It should be accepted after revision. 

My Comments and Suggestions to Authors:

1-The abstract is not convincing and is disorganized, it should be refined to precisely illustrate what authors have done in this paper within 200 words. The abstract must be a concise yet comprehensive reflection of what is in your paper. Remember that reader want to know:  1-what is the problem. 2- why the problem is relevant 3- wants an overview of your approach. 4-need to know the results.

1-Manuscript needs a good introduction, the introduction section of the manuscript is weak, authors are advised to improvise the introduction section.

2-In the Introduction part, the new features of the proposed method and the main advantages of the results over others should be clearly described.

3-An introduction should clearly highlight the motivation, problem statement, the objective of the paper, gap in the existing research and the novelty of the conducted research.

4-This application topic has not received much attention in the literature. However, the study, literature review and presentation require substantial improvement in several respects.

5. When I checked the results, I noticed that there were mistakes, please recheck.

6. I suggest extending the conclusions section to focus on the results you get, the method you propose, and their significance.

7. The article need to be revised with more experimentation by comparison relevant approach and algorithms.

8.   Expand literature review and cite most related work:

Tuple Linguistic Fermatean Fuzzy Decision-Making Method Based on COCOSO with CRITIC for Drip Irrigation System Analysis. Journal of Computational and Cognitive Engineering. https://doi.org/10.47852/bonviewJCCE2202356

Extended CODAS method for multi-attribute group decision-making based on 2-tuple linguistic Fermatean fuzzy Hamacher aggregation operators. Granul. Comput. (2022). https://doi.org/10.1007/s41066-022-00332-3

Assessment of Hydropower Plants in Pakistan: Muirhead Mean-Based 2-Tuple Linguistic T-spherical Fuzzy Model Combining SWARA with COPRAS. Arab J Sci Eng (2022). https://doi.org/10.1007/s13369-022-07081-0

Extending COPRAS Method with Linguistic Fermatean Fuzzy Sets and Hamy Mean Operators, Journal of Mathematics, Volume 2022 | Article ID 8239263 | https://doi.org/10.1155/2022/8239263.

Author Response

(The authors gave the same response as above.)

Reviewer 4 Report

After reviewing the article, I consider that it provides the necessary scientific arguments to draw a clear and precise conclusion from the article, which is detailed in the conclusions. In addition, after reviewing the research, the proposed measures are reliable and validate the hypothesis proposed, where it is verified that the model proposed by the authors is better than the traditional one when detecting the causes of failures related to personnel and machine performance. As a whole, and after a thorough analysis of it, I consider that it is suitable for publication in its current format.

Author Response

Dear Editors and Reviewers,

We would like to thank you very much for your patience and we appreciate the editors and reviewer for your valuable comments and suggestions on our manuscript entitled “Improving quality of services relevant with medical examination through DEMATEL-based Failure Mode and Effect Analysis”. We have studied the comments and suggestions carefully and have made the revisions which hopefully meet with expectations.

Round 2

Reviewer 3 Report

I accept the paper.